# Assess the efficacy of China's Inter-provincial Government Services policy: A quantitative evaluation based on PMC-Index model

Rong-qing Geng[1], Jian Wu[2]*

1 Yantai Institute, China Agricultural University, Yantai, Shandong, China, 2 School of Health Policy & Management, Nanjing Medical University, Nanjing, Jiangsu, China

* wujian0304@njmu.edu.cn

## Abstract

To face with the challenges of governance in the digital age, such as insufficient coordination between regional governments and low quality and efficiency of government services, China has proposed the Inter-provincial Government Services Policy. The policy is capable of realizing the government's ability to handle service matters for the public in different regions, thus facilitating the regional government's coordination and upgrading the level of government services. This paper collects and organizes the texts of 28 Inter-provincial Government Services policies, and uses ROSTCM6 text mining software to screen and identify the policy text content. Then a quantitative evaluation method based on the PMC model is proposed to examine the consistency and efficacy of the policy in this paper. The results show that: (a) The policy design is generally considered to be rational, with the majority of policies rated as excellent and a few rated as acceptable. There are no policies considered bad or perfect. From a certain point of view, these policies show obvious advantages in terms of policy nature, policy content and policy function. (b) The equilibrium of various policy indicators implies a high level of policy consistency. It indicates the overall coherence and coordination of the policies, contributing to enhanced predictability, credibility, and operationalization of policies, thereby establishing the foundation for their effective implementation. (c) There are still several weak points with the current policies, including the narrow scope of areas, the lack of medium- and long-term planning, and the insufficiently scientific nature of the instruments, evaluations and citations. This paper presents optimization recommendations aimed at addressing the aforementioned issues, which include expanding the scope of the policy, bolstering the long-term impact of the policy, and enhancing the quality of decision-making.

## 1. Introduction

With the advent of the digital age, governments around the world have undertaken digital transformation in the field of government services [1]. The digital transformation of government facilitates the process of data-driven decision-making and improves the efficiency of

**Data Availability Statement:** All the data are based on the texts of 28 Inter-provincial Government Services policies issued by China from 2020 to 2023. These policy documents were downloaded

by the authors from Chinese local government websites. And it is shown in file called "supporting information".

**Funding:** This work was supported by National Social Science Foundation of China (No. 23CSH057), Hebei Natural Science Foundation (No. G2023203015).The funders had no role in study design, data collection and analysis, decision to publish, or preparation of the manuscript.

**Competing interests:** The authors have declared that no competing interests exist.

government services. However, it has also brought unprecedented challenges to government governance [2]. Particularly, departmental and hierarchical boundaries in government constrain data sharing, resulting in persistent problems for governments in facilitating cross-regional public services. In response to these challenges, some countries, exemplified by the United Kingdom, have implemented "one-stop" government project [3]. And, China has also proposed Inter-provincial Government Services Policy [4] in 2020. It is the expansion and extension of "one-stop" government project. As an inter-provincial government service model, China's Inter-provincial Government Services Policy is different from the "one-stop" government project. It not only involves the coordination of services between government departments, but also involves governmental collaboration between different regions [5,6].

Since the implementation of the Inter-provincial Government Services Policy, numerous favorable impacts have emerged. This policy has addressed the issue of government service accessibility for individuals across different regions [7–9]. It has dismantled regional boundaries among governments, thereby fostering data sharing among regional authorities. So, how does the consistency and effectiveness of this policy fare? There is still a lack of an accurate and scientific evaluation to explore the strengths and weaknesses of the policy. This is not conducive to the long-term implementation[10]. Therefore, it is of great significance to evaluate the existing policy texts on Inter-provincial Government Services to gain deeper insights of their content.

In this paper, based on the texts of 28 Inter-provincial Government Services policies issued by China from 2020 to 2023, ROSTCM6 text mining software is utilized in screening and identifying the textual content of the policies, and a PMC (Policy Model Consistency) index model was constructed to quantitatively evaluate the different policies. The PMC index model evaluation method has the characteristics and advantages of systematic, holistic and predictive. By constructing a multi-dimensional comprehensive index system, it is able to conduct a comprehensive and systematic evaluation of the policy text. The trend of policy content can also be revealed through the analysis of historical data, which helps to identify potential problems and risks in advance and provides a basis for making forward-looking decisions.

The highlights of this paper are summarized as follows: (1) Theoretical Contributions: The first is the innovation of policy evaluation methods. This paper proposes a policy evaluation method based on the PMC model, which provides a new perspective for understanding the effect of policies. Through the quantitative study of the policy text, the lack of evaluation methods of the existing cross-provincial general policy is supplemented. With the help of the PMC index model, the PMC surface diagram is combined with the depression index to analyze the characteristics and shortcomings of the existing policies. This opens up new avenues for public policy evaluation. The second is to deepen the theory of policy evaluation. From the perspective of policy formulation, this paper conducts a quantitative text analysis of the inter-provincial general policy, conducts policy evaluation, and explains the effectiveness and consistency of policy implementation. (2) Practical implications: Firstly, quantitatively evaluate the effect of the policy. The quantitative evaluation of China's Inter-provincial Government Service Policy through the PMC index model can show the effect of policy implementation more intuitively. This approach can provide policymakers with strong data support. The second is to guide the direction of policy optimization. The PMC index model can not only evaluate the effectiveness of the strategy, but also visually show the strengths and weaknesses of each dimension of the strategy through the PMC surface diagram. There is still the possibility of policy optimization in terms of scope of application, timeliness and scientificity, which provides a clear direction for policy optimization for policymakers.

The rest of the paper is organized as follows: the Literature review section reviews the relevant literature. The Research design section describes the specific research design. The results of the quantitative evaluation of the policies are shown in Evaluation and Comparative

Analysis of Inter-provincial Government Services Policies section, and the Conclusions and Policy Implications section presents the conclusions and policy insights.

## 2. Literature review

In this section, we review the two types of literature that are most relevant to this study, namely, the research on the Inter-provincial Government Services Policy and the research on policy evaluation methods.

### 2.1 Research on Inter-provincial Government Services policy

At present, scholars' research on the Inter-provincial Government Services Policy mainly focuses on three aspects: the policy context, the policy background, and the policy practices.

1. Policy context: The design of the Inter-provincial Government Services Policy is a field of application for the digital transformation of government. Specifically, this policy emphasizes the digitalization of government operations at the technical level [11,12]. Scholars refine the concept of 'public-technical governance-overlapping action' and propose three practice modes for cross-domain governance of government services: vertically embedded, internally originated, and governance of government and society. They constructed the development path of cross-domain governance of government services in the four dimensions of strategy, supply and demand, technology, and ecology development path [13,14]. In terms of the development direction of cross-regional synergy of digital transformation of government services, the expectation of all countries is to provide e-services to enterprises and the public at any time and in any place through various means [15].

2. Policy background: Some scholars have pointed out that the local "one-network co-operation" is bounded by administrative divisions, and faces the problems of "power boundaries" and "differences in goals", which requires the data sharing mechanism to change from authoritative order to collaborative governance [16]. At the same time, the issue of cross-regional synergy is not a new proposition in the study of administration, on the one hand, there are a large number of theoretical discussions laying the cognitive foundation of cross-regional synergy [17–19], on the other hand, in the fields of regional economy [20], environmental governance [21–23], emergency management [24] and so on [25,26], it has already produced a rich empirical studies.

3. Policy practice: Some scholars have taken advanced demonstration zones as the starting point for case studies, explored the practice path of demonstration zones, explored the policy implementation dilemmas at the organizational level, technological level, and process level [27],or constructed an "institutional-technological-structural" framework for the implementation of "cross-provincial common office" in specific areas. The problems and challenges in the implementation process of inter-provincial government services are analyzed [28].

The above studies provide literature support for this paper to conduct research on Inter-provincial Government Services Policy, but the existing studies do not consider the effect of the policy text.

### 2.2 Research for policy evaluation

Policy evaluation is a complex and systematic project aimed at measuring and evaluating policy programs and providing a basis for policy formulation, adjustment and optimization [29]. And the key to conducting effective policy assessment lies in obtaining sufficient policy

information and using appropriate assessment tools [30]. With the in-depth and continuous development of policy evaluation practice, the methods of policy evaluation have become increasingly diversified, ranging from qualitative studies based on cases or expert reviews [31,32] to quantitative studies based on mathematical models [33]. Comprehensive evaluation methods based on empirical analysis, such as text data mining [34,35], social network analysis [36,37], fuzzy comprehensive evaluation [38,39], etc., have been developed. Commonly used empirical analysis tools are PSM-DID model analysis, instrumental variables and integrated control [40]. In recent years, some scholars have used PMC model to quantitatively evaluate manufacturing policies [41]. Text mining is the process of exploring the intrinsic patterns and connections from a large amount of unstructured textual information and mining valuable information from it. The Policy Modeling Research Consistency Index (PMC-Index) is a quantitative policy evaluation analysis method to evaluate the strengths and weaknesses of policy texts. The index can not only analyze the internal heterogeneity and the level of strengths and weaknesses of a policy in multiple dimensions, but also visualize the strengths and weaknesses of each dimension of the policy through the surface diagram, which provides a new way of thinking and a new perspective for the quantitative evaluation of policies [42,43].

In summary, there is a considerable body of literature on the policy of Inter-provincial Government Services and policy evaluation. Although some scholars have conducted studies on the Inter-provincial Government Services policy and provided valuable insights for this paper, these studies mainly focus on the policy implementation results. There are few studies on quantitative evaluation of the policy text from the perspective of policy formulation. However, the quality of Inter-provincial Government Services Policy is directly related to the effectiveness of their implementation. Although China has introduced various policy texts on Inter-provincial Government Services, their quality is still unknown. Therefore, this paper combines the text mining method and the PMC index model to quantitatively evaluate the collected Inter-provincial Government Service Policy texts. On the one hand, it aims to reveal the quality of Inter-provincial Government Service Policies, with a view to identifying the advantages and disadvantages of each policy. On the other hand, it also provides inspiration for the formulation and improvement of a new round of Inter-provincial Government Service Policies.

## 3. Research design

### 3.1 Research sample and methodology: Selection criteria and data collection

Under the background of China's Inter-provincial Government Services Policy, many provinces have successively introduced special policies for cross-provincial government service construction from 2020 to 2023.This provides an important way to change government functions and improve the capacity of government services. This paper reviews the policy documents and searches for the construction of inter-provincial government services. Twenty-eight policy documents were collected for analysis in this paper (see Table 1) by reviewing the policy documents and searching for keywords related to the inter-provincial government services policy. These policies are numbered as $P_1$-$P_{28}$ according to the order of their promulgation. The policy texts in this paper are taken from China's State Council Policy Document Library and the official portals of provincial governments to ensure their authority. The collection strategy is as follows:

Firstly, the State Council Policy Document Library contains all the normative policy documents that have been publicly released by the State Council, and the official provincial government portals are the most influential service platforms in each province. Secondly, in terms of searching subject terms, we choose inter-provincial government services as the keyword of

**Table 1. Inter-provincial government services policy texts.**

| Code | Policy Text Name | Date Issued |
|------|------------------|-------------|
| $P_1$ | Guiding Opinions of the General Office of the State Council on Accelerating the Promotion of Inter-provincial Government Services | Sept. 24, 2020 |
| $P_2$ | Notice of the General Office of the People's Government of Chongqing Municipality on the Issuance of the Work Program of Chongqing Municipality for Accelerating the Promotion of Inter-provincial Government Services | Nov. 18, 2020 |
| $P_3$ | Notice of the General Office of the People's Government of Gansu Province on the issuance of the Work Program of Gansu Province to promote high-frequency administrative service matters Inter-provincial Government Services | Nov. 19, 2020 |
| $P_4$ | Notice of the General Office of the People's Government of Guizhou Province on the Issuance of the Work Program of Guizhou Province for Accelerating the Promotion of Inter-provincial Government Services | Nov. 26, 2020 |
| $P_5$ | Notice of the General Office of the People's Government of Tianjin Municipality on the Issuance of the Work Program of Tianjin Municipality for Accelerating the Promotion of Inter-provincial Government Services | Nov. 27, 2020 |
| $P_6$ | Notice of the General Office of the People's Government of Guangdong Province on the Issuance of the Work Program of Guangdong Province for Accelerating the Promotion of Inter-provincial and Intra-provincial Government Services | Nov. 27, 2020 |
| $P_7$ | Notice of the General Office of the People's Government of Yunnan Province on the issuance of the Work Program of Yunnan Province for Accelerating the Promotion of Inter-provincial Government Services | Nov. 27, 2020 |
| $P_8$ | Notice of the General Office of the People's Government of Fujian Province on the issuance of the Work Program of Fujian Province for Accelerating the Promotion of Inter-provincial and Intra-provincial Government Services | Dec. 7, 2020 |
| $P_9$ | Notice of the General Office of the People's Government of Jiangxi Province on the issuance of the Work Program of Jiangxi Province for Accelerating the Promotion of Inter-provincial and Intra-provincial Government Services | Dec. 9, 2020 |
| $P_{10}$ | Notice of the General Office of the People's Government of Hebei Province on the issuance of the Work Program of Hebei Province for Accelerating the Promotion of Inter-provincial Government Services | Dec. 10, 2020 |
| $P_{11}$ | Notice of the General Office of the People's Government of Sichuan Province on the issuance of the Work Program of Sichuan Province for Accelerating the Promotion of Inter-provincial Government Services | Dec. 14, 2020 |
| $P_{12}$ | Notice of the General Office of the People's Government of Xinjiang Uygur Autonomous Region on the issuance of the Work Program of Xinjiang Uygur Autonomous Region for Accelerating the Promotion of Inter-provincial Government Services | Dec. 15, 2020 |
| $P_{13}$ | Notice of the General Office of the People's Government of the Inner Mongolia Autonomous Region on the issuance of the Work Program of the Inner Mongolia Autonomous Region for Accelerating the Promotion of Inter-provincial Government Services | Dec. 17, 2020 |
| $P_{14}$ | Notice of the General Office of the People's Government of Jiangsu Province on the issuance of the Work Program of Jiangsu Province for Accelerating the Promotion of Inter-provincial and Intra-provincial Government Services | Dec 28, 2020 |
| $P_{15}$ | Province on the issuance of the Work Program of Zhejiang Province for Accelerating the Promotion of Inter-provincial and Intra-provincial Government Services | Dec 29, 2020 |
| $P_{16}$ | Notice of the General Office of the People's Government of Hubei Province on the issuance of the Work Program of Hubei Province for Accelerating the Promotion of Inter-provincial Government Services | Dec 31, 2020 |
| $P_{17}$ | Notice of the General Office of the People's Government of Jilin Province on the issuance of the Work Program of Jilin Province for Accelerating the Promotion of Inter-provincial Government Services | Jan 4, 2021 |
| $P_{18}$ | Notice of the General Office of the People's Government of Anhui Province on the issuance of the Work Program of Anhui Province for Accelerating the Promotion of Inter-provincial Government Services | Jan 5, 2021 |
| $P_{19}$ | Notice of the General Office of the People's Government of Hunan Province on the issuance of the Work Program of Hunan Province for Accelerating the Promotion of Inter-provincial Government Services | Jan 19, 2021 |

(*Continued*)

**Table 1.** (Continued)

| Code | Policy Text Name | Date Issued |
|------|------------------|-------------|
| $P_{20}$ | Notice of the General Office of the People's Government of Guangxi Province on the issuance of the Work Program of Guangxi Province for Accelerating the Promotion of Inter-provincial Government Services | Mar 22, 2021 |
| $P_{21}$ | Notice of the General Office of the People's Government of Shandong Province on the issuance of the Work Program of Shandong Province for Accelerating the Promotion of Inter-provincial and Intra-provincial Government Services | Apr 1, 2021 |
| $P_{22}$ | Notice of the General Office of the People's Government of Tibet Autonomous Region on the issuance of the Work Program of Tibet Autonomous Region for Accelerating the Promotion of Inter-provincial Government Services in Five Provinces in Southwest China | Jul 6, 2021 |
| $P_{23}$ | Notice of the General Office of the People's Government of Shanxi Province on the issuance of the Work Program of Shanxi Province for Accelerating the Promotion of Inter-provincial and Intra-provincial Government Services | Jul 16, 2021 |
| $P_{24}$ | Notice of the General Office of the People's Government of Henan Province on the issuance of the Work Program of Henan Province for Accelerating the Promotion of Inter-provincial Government Services | Aug 6, 2021 |
| $P_{25}$ | Notice of the General Office of the People's Government of Liaoning Province on the issuance of the Work Program of Liaoning Province for Accelerating the Promotion of Inter-provincial, Intra-provincial and Regional Government Services | Aug 16, 2021 |
| $P_{26}$ | Notice of the General Office of the People's Government of Heilongjiang Province on the issuance of the Work Program of Heilongjiang Province for Accelerating the Promotion of Inter-provincial and Intra-provincial Government Services | Nov 17, 2021 |
| $P_{27}$ | Notice of the General Office of the People's Government of Qinghai Province on the issuance of the Work Program of Qinghai Province for Accelerating the Promotion of Inter-provincial Government Services | Nov 24, 2022 |
| $P_{28}$ | Notice of Beijing Municipal Administration of Government Services on the Issuance of the Work Program of Beijing Municipality for Accelerating the Promotion of Inter-provincial Government Services | Sep 14, 2023 |

policy filtering to ensure its accuracy. Thirdly, in terms of search strategy, the study includes two filtering criteria: (1) The text subject is limited to the State Council and provincial governments, and does not involve policies or industry standards of government departments. (2) The text is directly related to the inter-provincial government services, which is currently being implemented. Finally, we obtained 28 highly relevant and effective policy texts.

## 3.2 Policy feature recognition: Text mining and data analysis

Following the acquisition of policy texts, the next step involves applying text mining methods to preprocess the policy text in preparation for the model construction. Firstly, the ROSTCM 6.0 software is utilizing in performing word division and word frequency statistics on the policy. Secondly, words that have no obvious effect on analyzing the policy features are eliminated, such as nouns like "matters, licenses, materials", adjectives like "application-oriented", and words like "promote, realize, do, strengthen, promote, enhance, build, and so on". Then, the verbs of tendency such as "advance, realize, do, strengthen, promote, enhance, build, establish, optimize, speed up, perfect" are also eliminated. Finally, the word classification results are arranged according to the word frequency in ascending order, and the top 60 high-frequency words are extracted to form a high-frequency word list (see Table 2). By analyzing the high-frequency words in the policy text, we can identify the following characteristics of the provincial Inter-provincial Government Services Policy:

1. From the perspective of policy design concepts, the provincial policy implements the guiding opinions of the State Council on accelerating the promotion of inter-provincial

**Table 2. Policy text key words' frequency statistics table.**

| Serial Number | Vocabulary | Frequency | Serial Number | Vocabulary | Frequency |
|---|---|---|---|---|---|
| 1 | Services | 2136 | 31 | Mechanisms | 156 |
| 2 | Government | 1708 | 32 | Docking | 152 |
| 3 | Departments | 682 | 33 | Reform | 143 |
| 4 | Processing | 590 | 34 | Window | 142 |
| 5 | Units | 501 | 35 | Demand | 140 |
| 6 | Platforms | 466 | 36 | Synergy | 135 |
| 7 | Business | 423 | 37 | Standards | 135 |
| 8 | Data | 360 | 38 | Applications | 124 |
| 9 | Sharing | 316 | 39 | Support | 124 |
| 10 | Crowd | 281 | 40 | National | 120 |
| 11 | Integration | 275 | 41 | Guidance | 120 |
| 12 | Enterprise | 266 | 42 | Capabilities | 119 |
| 13 | Electronic | 244 | 43 | Issues | 119 |
| 14 | Unified | 232 | 44 | Coordination | 118 |
| 15 | offsite | 226 | 45 | Centers | 115 |
| 16 | systems | 222 | 46 | Area | 114 |
| 17 | Various Locations | 202 | 47 | Online | 114 |
| 18 | management | 202 | 48 | Levels | 112 |
| 19 | acceptance | 200 | 49 | Mode | 111 |
| 20 | Provincial | 200 | 50 | Regulatory | 110 |
| 21 | List | 185 | 51 | Regulate | 106 |
| 22 | Application | 181 | 52 | Personnel | 105 |
| 23 | National | 174 | 53 | Responsibilities | 103 |
| 24 | Autonomous Region | 169 | 54 | Policies | 102 |
| 25 | Process | 167 | 55 | Time Frame | 100 |
| 26 | Regions | 167 | 56 | Organization | 100 |
| 27 | Cities | 165 | 57 | Administrations | 100 |
| 28 | Province | 163 | 58 | Rely on | 97 |
| 29 | Government | 163 | 59 | Comprehensive | 90 |
| 30 | Lead | 158 | 60 | Programs | 89 |

government services, adheres to the principle of "people-centered development" and adheres to the concept of new development concepts.

2. From the perspective of policy implementation subjects, inter-provincial government services not only need national, inter-regional and intra-regional coordination, but also need national integrated government service platforms. The national comprehensive government service platform and government service agencies at all levels play an important role in bridging the blockage of business chains and data sharing.

3. From the perspective of policy guarantee mechanism, policy guarantee mechanisms, including integration, management, acceptance, and coordination, are crucial for accelerating the construction of Inter-provincial Government Services.

Social network analysis is a set of norms and methods to analyze the structure and attributes of social relations. It mainly analyzes the structure and attributes of the relationships constituted by different social units [44]. The data analysis software *Gephi* is utilized in constructing

a semantic network based on high-frequency words, as shown in Fig 1. The proximity of each subject word is determined by the proximity of the nodes within the semantic network and the thickness of the connecting line, the thicker the straight line, the closer the relationship between two keywords. Also, the size of nodes represents the strength of centrality. If a node has more connections to other nodes, its degree centrality is stronger, which means that the node is more important.

The visualization results indicate that certain keywords, such as 'government' and 'services', hold significant centrality within the semantic network. "Government", "services" and "departments" are three main key words, indicates that government departments have the primary responsibility as the main body of governance, with the goal of optimizing the delivery of government services. "Sharing" and "platforms" are also an important concern, demonstrates the need to fully apply the government service platform for data sharing and data security protection. Through the keyword network analysis, the focus of the government's provision of

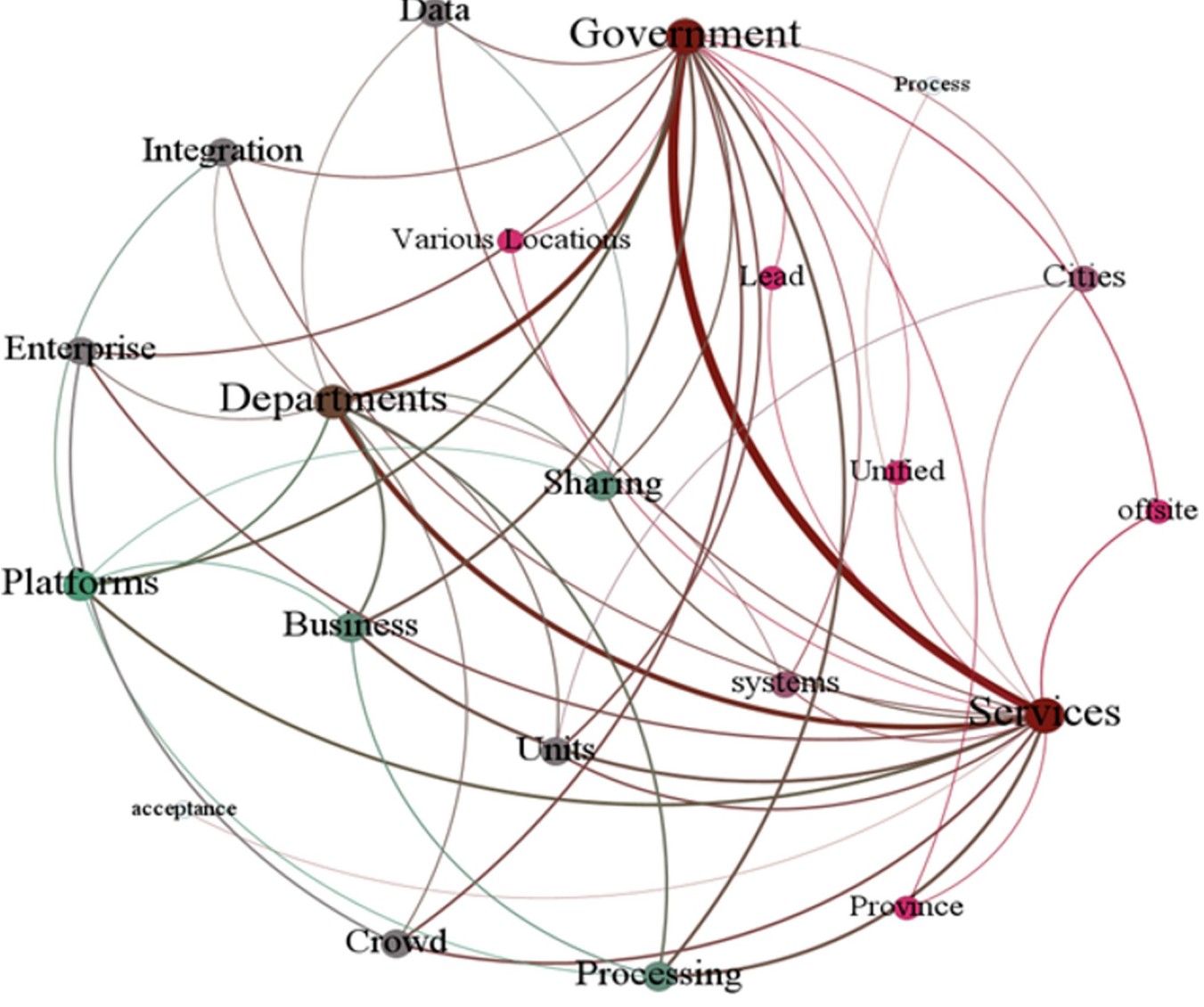

**Fig 1. Keyword network map.**

government services is clarified, namely, the three aspects of nature, subject and tools. This provides an important basis for the selection of policy evaluation variables and indicators for the construction of the PMC index model below. At the same time, the relationship between the keywords also illustrates the internal logic between the variables of the PMC index model, making the evaluation index system form an organic whole.

### 3.3 Construction of PMC index model

In this paper, the PMC index model is used to quantitatively evaluate the Inter-provincial Government Services policies issued by the state. The PMC index model is proposed by Estrada [45] based on the Omnia Mobilis hypothesis [46], which holds that everything in the world is moving and connected, and that all relevant variables should be considered as much as possible in the modeling and no relevant variables should be removed. The construction process consists of the following four steps: (1) classification of variables and identification of parameters; (2) creation of multi-input-output tables; (3) measurement of PMC indices; (4) construction of PMC surfaces [47].

**3.3.1 Variable classification and parameter identification.** Combined with Dai et.al.[1] and Song [48] research, this paper constructed an index system of Inter-provincial Government Services policy rating (see Table 3). The index system consists of 10 first-level variables and 35 second-level variables, among which the first-level variables are: policy nature($X_1$), policy subject($X_2$), policy object($X_3$), policy content($X_4$), policy function($X_5$), policy timeliness ($X_6$), policy domain($X_7$), policy instrument($X_8$), policy evaluation($X_9$), policy citation($X_{10}$). All the sub-variables of the remaining level 1 variables and their associated evaluation criteria are shown in Table 3. The tenth level 1 index ($X_{10}$) is policy citation, which describes whether the policy cites other policies. It has no sub-variables.

**3.3.2 Establish multi-input-output table to calculate first-level index.** As a basic analytical framework for quantifying the Inter-provincial Government Services policy, the multi-input-output table can measure any single variable in the evaluation index system by storing a large amount of data, which is a key part of the PMC index model construction. So, a multi-input-output table based on the combination of new smart city policy variable classification and parameter identification is established, which is shown in Table 4. In addition, in order to further enhance the quality and reliability of the policy evaluation results, in the process of assigning values to the secondary variables based on Table 3, the corresponding variables can only take the value of 1 if the text of the policy to be evaluated is able to describe the content of the secondary variables in a very clear way; for the content that is difficult to be determined, a Delphi method is used in determining the specific values of the variables, which is able to reduce the subjective error of the policy evaluation to a certain extent.

**3.3.3 Calculation of PMC index.** Besides the above first-level and second-level indexes, two additional parameters were introduced into the structure of the PMC index. If the second-level index could fit into the policy model, it was denoted by "1"; if the second-level index could not fit into the policy model, it was denoted by "0". That is, each parameter was coded to the binary values "0" or "1". Without losing the generality, this article will give equal weights to all the secondary indicators. The first-level indexes were calculated via summarizing all the secondary indexes using Eq (3), and then the final PMC index was obtained by summing up the values of all variables using Eq (4). Finally, the surface graph, to display the resulting PMC index matrix more intuitively, is drawn. The calculation formulas for the PMC index are

**Table 3. PMC evaluation index system and evaluation standard.**

| First-Level Index | Second-Level Index | Evaluation Criteria |
| --- | --- | --- |
| $X_1$ | Forecasting $X_{11}$<br>Regulation $X_{12}$<br>Recommendation $X_{13}$<br>Guidance $X_{14}$ | Policies are predictive, yes 1, no 0<br>Policy is regulatory in nature, yes 1, no 0<br>Policy is recommendatory, yes 1, no 0<br>Policy is guiding, yes 1, no 0 |
| $X_2$ | Government Departments $X_{21}$<br>Specialized Agencies $X_{22}$<br>Technology Company $X_{23}$<br>Enterprises, Citizens $X_{24}$ | Whether the policy subject involves the government and various functional departments, yes 1, no 0<br>Whether the main body of the policy involves experts, think tanks and research institutions, yes 1, no 0<br>Whether the main body of the policy involves technology companies and local agencies, yes 1, no 0<br>Whether the policy subject involves business enterprises and private citizens, yes 1, no 0 |
| $X_3$ | Enterprises $X_{31}$<br>Ordinary Citizen $X_{32}$<br>Institutions of higher learning $X_{33}$<br>Research Organization $X_{34}$<br>Other Organizations $X_{35}$ | Whether the policy object involves enterprises and legal persons, yes 1, no 0<br>Whether the policy object involves the public, citizens, yes 1, no 0<br>Whether the policy object involves institutions, colleges and universities, yes 1, no 0<br>Whether the policy object involves scientific research institutions, research institutes, yes 1, no 0<br>Whether the policy object involves social organizations, institutions, etc., yes 1, no 0 |
| $X_4$ | Off-site Processing $X_{41}$<br>Regional cooperation $X_{42}$<br>Online Integration $X_{43}$<br>Offline synergy $X_{44}$ | Whether the policy involves cross-location processing of matters, yes 1, no 0<br>Whether the policy involves inter-regional exchanges and cooperation, yes 1, no 0<br>Whether the policy involves the integration of online service portals, yes 1, no 0<br>Whether the policy involves synergizing offline service mechanisms, yes 1, no 0 |
| $X_5$ | Optimize Governance $X_{51}$<br>Promoting Development $X_{52}$<br>Improving People's Livelihoods $X_{53}$<br>Enhancing data sharing$X_{54}$ | Whether the policy has the function of optimizing government governance, yes 1, no 0<br>Whether the policy has the function of promoting economic development, yes 1, no 0<br>Whether the policy has the function of improving people's livelihood services, yes 1, no 0<br>Whether the policy has the function of enhancing data sharing, yes 1, no 0 |
| $X_6$ | Short-term $X_{61}$<br>Medium-term $X_{62}$<br>Long-term $X_{63}$ | Whether the policy involves content within 3 years, yes 1, no 0<br>Whether the policy is related to the content of 3–5 years, yes 1, no 0<br>Whether the policy covers more than 5 years, yes 1, no 0 |
| $X_7$ | Economic $X_{71}$<br>Social $X_{72}$<br>Science and Technology $X_{73}$<br>Political $X_{74}$<br>Environment $X_{75}$ | Whether the policy covers economic areas, yes 1, no 0<br>Whether the policy covers social areas, yes 1, no 0<br>Whether the policy covers scientific and technological areas, yes 1, no 0<br>Whether the policy covers the political domain, yes 1, no 0<br>Whether the policy involves the environmental domain, yes 1, no 0 |
| $X_8$ | Organization and Coordination $X_{81}$<br>Rule of Law System $X_{82}$<br>Supervision and Evaluation $X_{83}$<br>Publicity and guidance $X_{84}$ | Whether the policy involves organizational leadership and coordination, yes 1, no 0<br>Whether the policy involves rule of law safeguards and institutional support, yes 1, no 0<br>Whether the policy involves supervision, guidance and service evaluation, yes 1, no 0<br>Whether the policy involves publicity and promotion and interpretation guidance, yes 1, no 0 |
| $X_9$ | Adequate basis $X_{91}$<br>Informative planning $X_{92}$<br>Geographically distinctive $X_{93}$ | Whether the policy is based on sufficient grounds, yes 1, no 0<br>Whether the policy program is specific and feasible, yes 1, no 0<br>Whether the policy has distinctive regional characteristics, yes 1, no 0 |
| $X_{10}$ | / | Whether the policy cites other policy documents, yes 1, no 0 |

detailed below:

$$X \sim N[0, 1] \tag{1}$$

$$X = \{XR : [0\sim1]\} \tag{2}$$

$$X_t = \left( \sum_{j=1}^{4} \frac{X_{tj}}{T\left(X_{tj}\right)} \right), t = 1, 2, \cdots, 10 \tag{3}$$

**Table 4. Multi-input-output table.**

| First-Level Index | Secondary Index |
|---|---|
| $X_1$ | $X_{11}, X_{12}, X_{13}, X_{14}$ |
| $X_2$ | $X_{21}, X_{22}, X_{23}, X_{24}$ |
| $X_3$ | $X_{31}, X_{32}, X_{33}, X_{34}, X_{35}$ |
| $X_4$ | $X_{41}, X_{42}, X_{43}, X_{44}$ |
| $X_5$ | $X_{51}, X_{52}, X_{53}$ |
| $X_6$ | $X_{61}, X_{62}, X_{63}$ |
| $X_7$ | $X_{71}, X_{72}, X_{73}, X_{74}, X_{75}$ |
| $X_8$ | $X_{81}, X_{82}, X_{83}, X_{84}$ |
| $X_9$ | $X_{91}, X_{92}, X_{93}$ |
| $X_{10}$ | / |

$$PMC = X_1\left(\sum_{i=1}^{4}\frac{X_{1i}}{4}\right) + X_2\left(\sum_{j=1}^{4}\frac{X_{2j}}{4}\right) + X_3\left(\sum_{k=1}^{5}\frac{X_{3k}}{5}\right) + X_4\left(\sum_{l=1}^{4}\frac{X_{4l}}{4}\right)$$
$$+ X_5\left(\sum_{m=1}^{3}\frac{X_{5m}}{3}\right) + X_6\left(\sum_{n=1}^{3}\frac{X_{6n}}{3}\right) + X_7\left(\sum_{o=1}^{5}\frac{X_{7o}}{5}\right) + X_8\left(\sum_{p=1}^{4}\frac{X_{8p}}{4}\right)$$
$$+ X_9\left(\sum_{q=1}^{3}\frac{X_{9q}}{3}\right) + X_{10} \tag{4}$$

According to the calculation results of the PMC index and the research of Estrada [45], the policy evaluation of the PMC index can be divided into four grades (see Table 5). Specifically, if the PMC index value was less than 4.99, the policy was regarded as having "bad consistency"; if the PMC index value was between 5.00 and 6.99, the policy was regarded as having "acceptable consistency"; if the PMC index value was between 7.00 and 8.99, the policy was regarded as having "excellent consistency"; and if the PMC index value was between 9.00 and 10.00, the policy was regarded as having "perfect consistency".

**3.3.4 PMC surface drawing.** The PMC surface figures (Figs 2 and 3) display a symmetric surface representing the values of nine first-level indexes, providing insights into the strengths and weaknesses of Inter-provincial Government Services policies. Since the values of policy nature ($X_1$) of 28 Inter-provincial Government Services policies are all 1, the policy nature index variables are eliminated, and a $3 \times 3$ matrix is constructed according to the values of the remaining indicators (see Eq (5)). Due to the limitation of space, this paper only presents the PMC surface graphs of the policies with the highest and the lowest scores of the PMC indexes, i.e., $P_{18}$($P_{25}$ same value as $P_{18}$) and $P_{19}$. Different colors in the PMC surface represent different index values, and the convex part of the surface indicates higher scores for the policy-level variables, while the concave part indicates lower scores for the policy-level variables (see Figs 2 and 3 for details).

$$PMC = \begin{bmatrix} X_2 & X_3 & X_4 \\ X_5 & X_6 & X_7 \\ X_8 & X_9 & X_{10} \end{bmatrix} \tag{5}$$

**Table 5. Classification of policy evaluation grades.**

| PMC Index | 0–4.99 | 5.00–6.99 | 7.00–8.99 | 9.00–10 |
|---|---|---|---|---|
| Policy Consistency | Bad | acceptable | excellent | perfect |

## 4. Evaluation and comparative analysis of Inter-provincial Government Services policies

### 4.1 Calculation of PMC Index of Inter-provincial Government Services policy

The evaluated results are calculated by using Eqs (1)–(4), and the grade evaluation of 28 representative policies were conducted according to Table 5, then the final evaluation results of 28 Inter-provincial Government Services policies are obtained. The evaluation grade of 28 Inter-provincial Government Services policies is divided into 4 grades (see Table 6).

### 4.2 Quantitative evaluation analysis of the polices

**4.2.1 Overview of policies: Analyzing overall characteristics.** The PMC index scores of the 28 policies (see Table 6) ranged from 6.45 to 8.62, with a mean value of 7.71. Notably, policies 15, 16, and 19 attained acceptable levels, while the others achieved excellence. It indicates that the 28 policies are of relatively good quality, with a certain degree of science and rationality, and can provide guidance for the construction of Inter-provincial Government Services. In terms of the PMC index ranking of the policy samples, $P_{18} = P_{25} > P_{23} > P_{10} > P_{26} > P_1$, reflecting

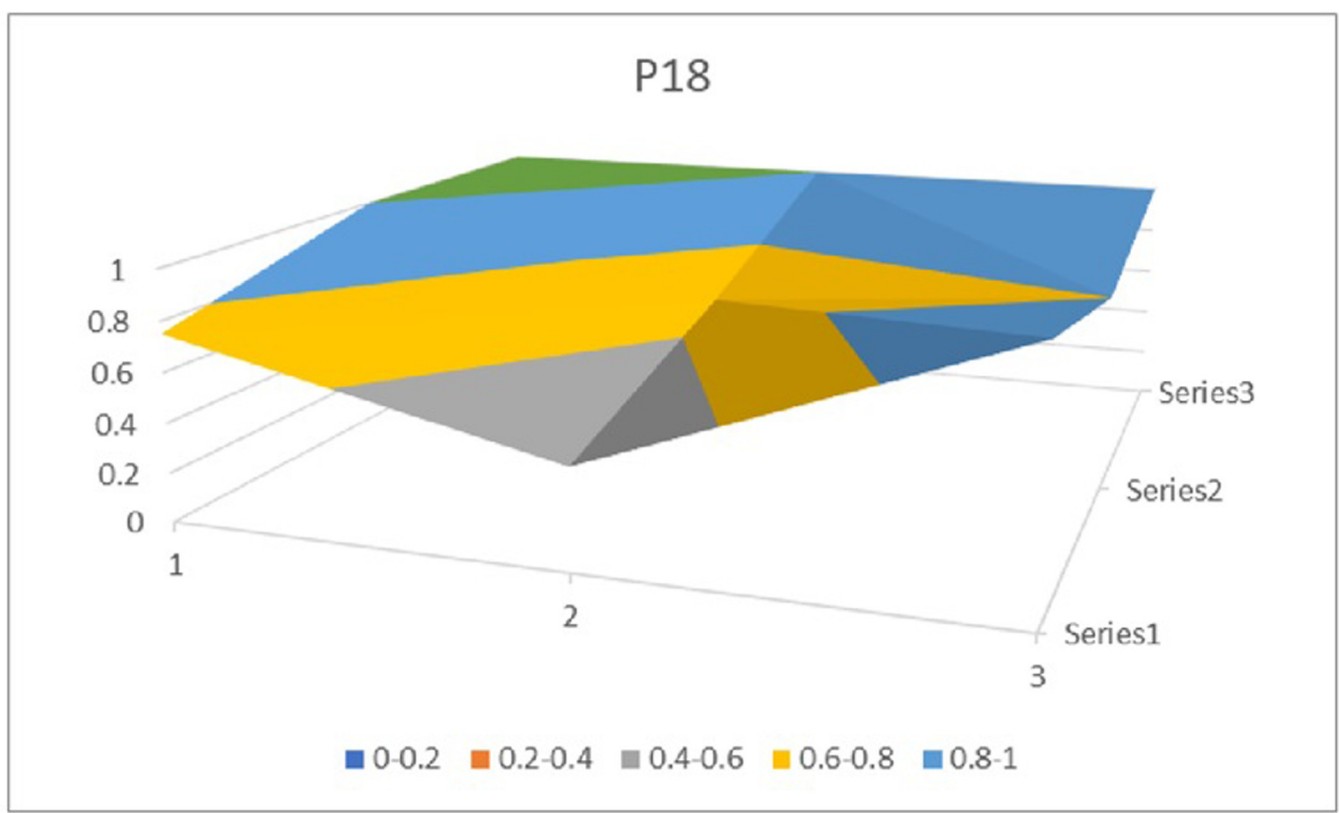

**Fig 2. PMC surface figure of P18.**

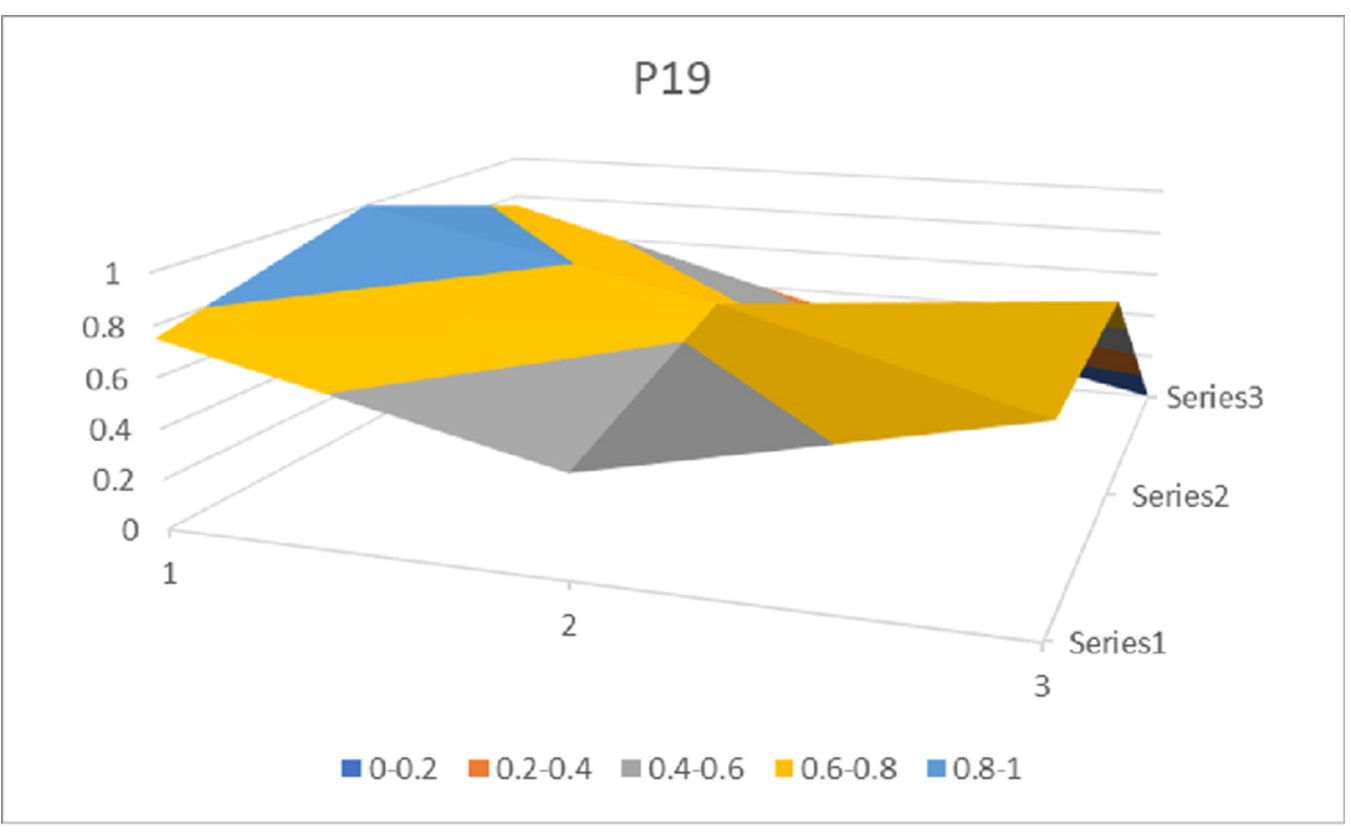

**Fig 3. PMC surface figure of P19.**

the tendency that China's Inter-provincial Government Services policies are "better at the bottom than at the top", *i.e.*, the lower the policy level, the better the policies promulgated by the government departments. That is, the lower the policy level, the higher the quality of the Inter-provincial Government Services policy. The main reason for this feature is that policies at the national level mainly play an overall and strategic guiding role, and many of their provisions are relatively general, whereas as the level of government decreases, there is a higher demand for operationalization of policies, i.e., in order to effectively promote the work of open government data in their jurisdictions, local governments focus on implementing the basic policies of higher levels of government to formulate more specific action plans and management methods. To effectively reveal the differences between the policies, the advantages and disadvantages of the policies can be analyzed by comparing the scores of each level of variables with the mean value and the differences between the "perfect" policies.

### 4.2.2 PMC surface diagram of policy average

To visually represent the PMC index calculation results, we transform the first-level index scores of each policy into a third-order matrix by using Eq (5), facilitating analysis of each policy's internal consistency level through surface concavity and convexity.

In the PMC surface diagram, the color of the blocks varies depending on the indicator scores. A convex surface indicates that a policy has a high score for the indicator, while a

**Table 6.  Calculation results of PMC index of 28 Inter-provincial Government Services policies.**

| Policy | $X_1$ | $X_2$ | $X_3$ | $X_4$ | $X_5$ | $X_6$ | $X_7$ | $X_8$ | $X_9$ | $X_{10}$ | PMC | Rank | Grade |
|---|---|---|---|---|---|---|---|---|---|---|---|---|---|
| $P_1$ | 1.00 | 0.75 | 0.60 | 1.00 | 1.00 | 0.33 | 0.80 | 1.00 | 0.67 | 1.00 | 8.15 | 6 | excellent |
| $P_2$ | 1.00 | 0.75 | 0.40 | 1.00 | 1.00 | 0.33 | 0.80 | 1.00 | 1.00 | 0.00 | 7.28 | 21 | excellent |
| $P_3$ | 1.00 | 0.75 | 0.40 | 1.00 | 1.00 | 0.33 | 0.80 | 1.00 | 1.00 | 0.00 | 7.28 | 21 | excellent |
| $P_4$ | 1.00 | 0.75 | 0.40 | 1.00 | 1.00 | 0.67 | 0.80 | 0.75 | 1.00 | 0.00 | 7.37 | 19 | excellent |
| $P_5$ | 1.00 | 0.75 | 0.40 | 0.75 | 1.00 | 0.67 | 0.80 | 0.75 | 1.00 | 1.00 | 8.12 | 7 | excellent |
| $P_6$ | 1.00 | 0.75 | 0.40 | 1.00 | 1.00 | 0.33 | 0.80 | 0.75 | 1.00 | 1.00 | 8.03 | 9 | excellent |
| $P_7$ | 1.00 | 0.75 | 0.40 | 0.75 | 1.00 | 0.33 | 0.80 | 1.00 | 1.00 | 1.00 | 8.03 | 9 | excellent |
| $P_8$ | 1.00 | 0.50 | 0.40 | 0.75 | 1.00 | 0.33 | 0.80 | 0.75 | 0.67 | 1.00 | 7.20 | 23 | excellent |
| $P_9$ | 1.00 | 0.50 | 0.40 | 1.00 | 1.00 | 0.33 | 0.80 | 0.75 | 1.00 | 1.00 | 7.78 | 15 | excellent |
| $P_{10}$ | 1.00 | 0.75 | 0.40 | 1.00 | 1.00 | 0.33 | 0.80 | 1.00 | 1.00 | 1.00 | 8.28 | 4 | excellent |
| $P_{11}$ | 1.00 | 0.75 | 0.40 | 0.75 | 1.00 | 0.33 | 0.80 | 1.00 | 1.00 | 1.00 | 8.03 | 9 | excellent |
| $P_{12}$ | 1.00 | 0.75 | 0.40 | 0.75 | 1.00 | 0.67 | 0.80 | 1.00 | 0.67 | 1.00 | 8.04 | 8 | excellent |
| $P_{13}$ | 1.00 | 0.75 | 0.40 | 0.75 | 1.00 | 0.33 | 0.80 | 1.00 | 1.00 | 1.00 | 8.03 | 9 | excellent |
| $P_{14}$ | 1.00 | 0.75 | 0.40 | 1.00 | 1.00 | 0.33 | 0.80 | 0.75 | 1.00 | 1.00 | 8.03 | 9 | excellent |
| $P_{15}$ | 1.00 | 0.75 | 0.40 | 0.75 | 1.00 | 0.33 | 0.80 | 0.75 | 1.00 | 0.00 | 6.78 | 26 | acceptable |
| $P_{16}$ | 1.00 | 0.75 | 0.40 | 0.75 | 1.00 | 0.33 | 0.80 | 0.50 | 1.00 | 0.00 | 6.53 | 27 | acceptable |
| $P_{17}$ | 1.00 | 0.75 | 0.40 | 1.00 | 1.00 | 0.67 | 0.80 | 0.75 | 1.00 | 0.00 | 7.37 | 19 | excellent |
| $P_{18}$ | 1.00 | 0.75 | 0.40 | 1.00 | 1.00 | 0.67 | 0.80 | 1.00 | 1.00 | 1.00 | 8.62 | 1 | excellent |
| $P_{19}$ | 1.00 | 0.75 | 0.40 | 0.75 | 1.00 | 0.67 | 0.80 | 0.75 | 0.33 | 0.00 | 6.45 | 28 | acceptable |
| $P_{20}$ | 1.00 | 0.75 | 0.40 | 1.00 | 1.00 | 0.33 | 0.80 | 0.75 | 1.00 | 1.00 | 8.03 | 9 | excellent |
| $P_{21}$ | 1.00 | 0.75 | 0.40 | 0.75 | 1.00 | 0.33 | 0.80 | 0.75 | 1.00 | 1.00 | 7.78 | 15 | excellent |
| $P_{22}$ | 1.00 | 0.75 | 0.40 | 0.75 | 1.00 | 0.33 | 1.00 | 0.50 | 1.00 | 1.00 | 7.73 | 17 | excellent |
| $P_{23}$ | 1.00 | 0.75 | 0.40 | 1.00 | 1.00 | 0.67 | 0.80 | 0.75 | 1.00 | 1.00 | 8.37 | 3 | excellent |
| $P_{24}$ | 1.00 | 0.75 | 0.40 | 1.00 | 1.00 | 0.67 | 0.80 | 1.00 | 1.00 | 0.00 | 7.62 | 18 | excellent |
| $P_{25}$ | 1.00 | 0.75 | 0.40 | 1.00 | 1.00 | 0.67 | 0.80 | 1.00 | 1.00 | 1.00 | 8.62 | 1 | excellent |
| $P_{26}$ | 1.00 | 0.75 | 0.40 | 0.75 | 1.00 | 0.33 | 1.00 | 1.00 | 1.00 | 1.00 | 8.23 | 5 | excellent |
| $P_{27}$ | 1.00 | 0.75 | 0.40 | 1.00 | 1.00 | 0.33 | 0.80 | 0.75 | 1.00 | 0.00 | 7.03 | 24 | excellent |
| $P_{28}$ | 1.00 | 0.75 | 0.40 | 1.00 | 0.75 | 0.33 | 0.80 | 0.75 | 1.00 | 0.00 | 6.78 | 25 | excellent |
| Average | 1.00 | 0.73 | 0.41 | 0.89 | 0.99 | 0.44 | 0.81 | 0.84 | 0.94 | 0.64 | 7.70 | / | / |

concave surface indicates that a policy has a low score for the indicator. Due to space limitation and to further demonstrate the overall consistency level of policies, this paper selects the PMC surface diagram of policy average.

As can be seen from the surface diagram, the overall structure of policies is more balanced. The smoother the surface is, the higher the policy consistency level is; the larger the surface concavity is, the lower the policy consistency level is. From the average PMC surface in Fig 4, we can see that the consistency level of China's current Inter-provincial Government Services Policy is relatively high, with a balanced force of various indicators and a reasonable structure.

## 4.3 Comparation analysis

**4.3.1 Comparison with the mean value.**   Upon comparing the scores of each policy variable with the mean value, it becomes evident that:

1. Since each policy has the characteristics of forecasting, regulating, advising and guiding, and the functions of the policy include optimizing government governance, promoting economic development and improving people's livelihood, the scores of the policy nature $X_1$ and the policy function $X_5$ are equal to the mean value, which are both 1.

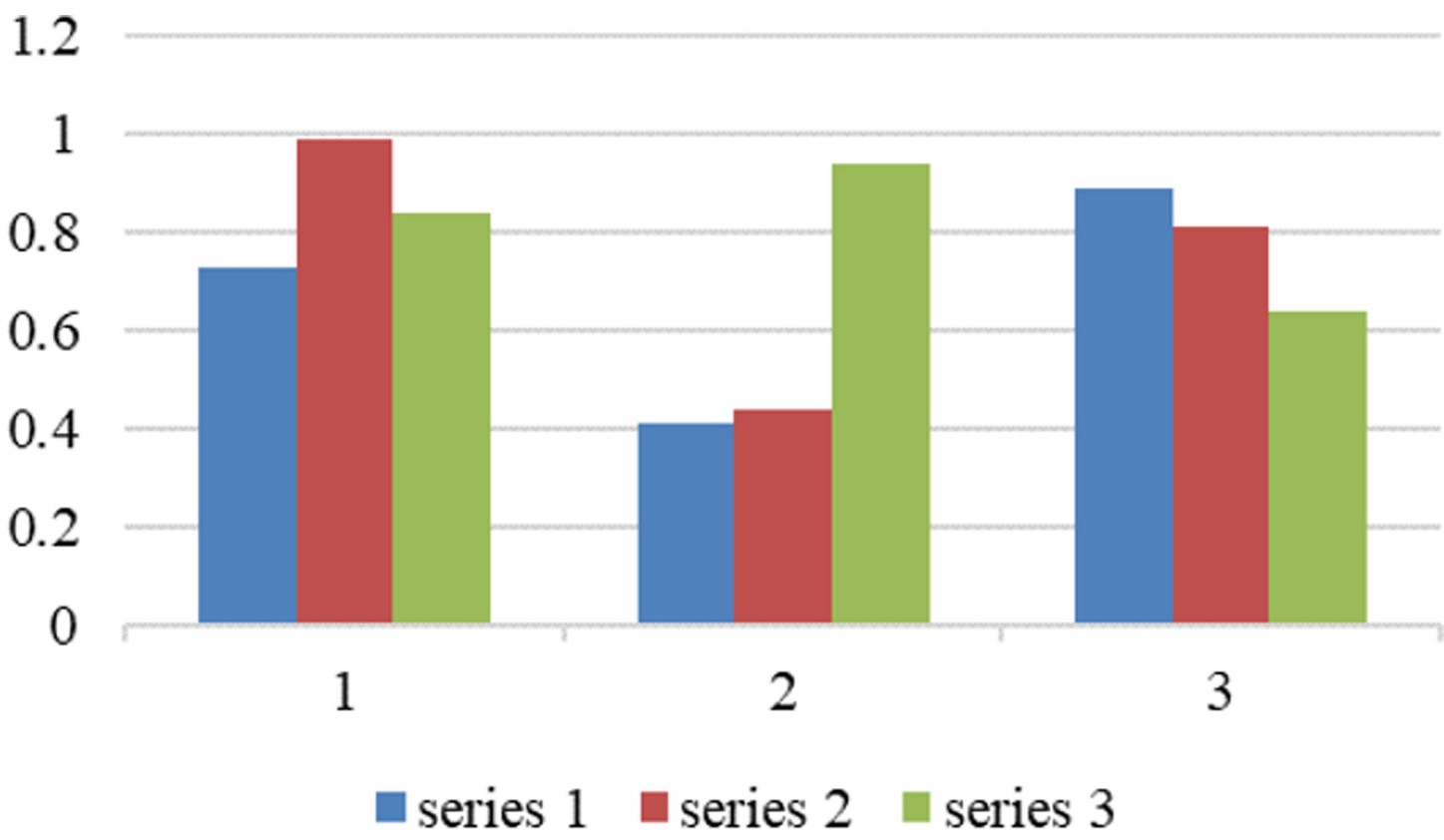

**Fig 4. PMC surface diagram of policy average.**

2. Two and four policies scored below the mean in the policy subject $X_2$ and the policy evaluation $X_9$ respectively, i.e. more than half of the policies scored above the mean. This shows that the main body of policy formulation includes the central government, provincial and municipal governments and their functional departments, technology companies, local agencies, as well as clerical enterprises and citizens, but all of them are involved in the advice of experts and research institutes; the policy evaluation in addition to the national central policy, the vast majority of provincial governments are able to take the central policy as a guide to the implementation of the actual situation in accordance with the local implementation and adjustments, while the central policy is mainly to play a role in guiding and advising the central policy. The central policy mainly plays the role of guidance and advice, and cannot fully take into account the different characteristics and needs of localities. From this, we can conclude the importance of the policy subject as the core component of the policy system, and the necessity for local policies to be optimized with distinctive regional characteristics.

3. More than half of the policies in Policy Object $X_3$, Policy Timeliness $X_6$, Policy Field $X_7$, and Policy Means $X_8$ scored lower than the average value, and among them, except for the central policy involving institutions, the rest of the policy objects only involve enterprises and ordinary citizens. This suggests that due to imperfect supporting institutions and technical constraints, the Inter-provincial Government Services policy is not well adapted to institutions of higher learning, scientific research institutions and other organizations; the policy's timeliness has not yet involved long-term and medium-term policy content,

meaning that the policy's timeliness is mainly short-term, and although it involves specific annual work objectives, it lacks a long-term deployment of more than five years and a medium-term development plan. In terms of policy areas, most of the policies focus on economic, social, scientific and technological, and political areas, but do not pay attention to the environmental area.

**4.3.2 Comparison and analysis with "Perfect" policies.** In order to better identify the optimization path of Inter-provincial Government Services policy, it is necessary to analyze the gap between each policy and the "perfect" policy. Therefore, this study draws on the practice of Song [48] and Du & Zhou [49], comparing the score of the first-level indicators with the "perfect" policy with full scores in all evaluation indicators, and the difference between the two is the concavity index (see Table 7).

1. In terms of the vertical dimension, i.e., the concavity of the indicators at each level of the policy. All policies have the greatest degree of concavity in Policy Direction $X_3$, with a mean value of 0.59. Analysis of the policy texts reveals that all 28 policies involve enterprises and ordinary citizens, and attach importance to the provision of convenient services to them, but are not specific about the benefits to institutions of higher education, scientific research institutes, and other organizations such as social groups and institutions. The mean value of the depression index of Policy Timeliness $X_6$ is 0.57, ranking second. It can be seen from the policy texts that the timeliness of most policies, including the central policies, is mainly short-term, lacks medium-term development plans and long-term deployments, and fails to make scientific and rational predictions of the future development direction and goals on the basis of grasping the current situation. The mean value of the concavity index of policy citation $X_{10}$ is 0.37. Most provincial and municipal policies, including central policies, generally cite Xi Jinping's thought on socialism with Chinese characteristics, the spirit of the 19th National Congress of the Party, and relevant laws and regulations, or use $P_1$ as a guideline, while ten policies do not make citations, which makes the policies lack a clear policy basis. The mean value of the concavity index of policy subject $X_2$ is 0.28. Specifically, the leading role of government departments and the importance of enterprises' and citizens' suggestions and policy participation are all taken into account in the process of participating in decision-making and policy implementation, so as to guarantee the smooth promotion of policy work "from the top to the bottom" and the realization of the goal of policies to meet the needs of the people, but all the policies ignore big data. However, all the policies have neglected the important role of big data experts, think tanks and research institutes as "advisors" in the operation of the policy system, while policies $P_8$ and $P_9$ have not included technology companies, agencies and other specialized institutions as partners, neglecting their role in providing professional support. In Policy Area $X_7$, all policies focus on economic, social, scientific and technological, and political services, but very few focus on the environment.

2. In terms of the horizontal dimples of policies, the dimples of 28 policies range from 1.38 to 3.55, with $P_{18}$ and $P_{25}$ having the lowest dimples, *i.e.*, closest to the "perfect policy", and $P_{19}$ having the largest gap between the "perfect policy" and the "perfect policy" in comparison to the other policies. The gap between $P_{19}$ and the "perfect policy" is the largest. In the case of $P_{19}$, as shown in Table 7, the gap between $P_{19}$ and the "perfect policy" is larger in the three aspects of policy object $X_3$, policy evaluation $X_9$, and policy citation $X_{10}$. Tracing back to the assignment of each secondary variable, it can be seen that the defects of $P_{14}$ in policy object $X_3$ are the biggest point of loss, but they are no different from other policies; in policy

**Table 7. Concavity indices for the 28 policies.**

|  | $X_1$ | $X_2$ | $X_3$ | $X_4$ | $X_5$ | $X_6$ | $X_7$ | $X_8$ | $X_9$ | $X_{10}$ | Concavity Index | Grade |
|---|---|---|---|---|---|---|---|---|---|---|---|---|
| $P_1$ | 0.00 | 0.25 | 0.40 | 0.00 | 0.00 | 0.67 | 0.20 | 0.00 | 0.33 | 0.00 | 1.85 | 23 |
| $P_2$ | 0.00 | 0.25 | 0.60 | 0.00 | 0.00 | 0.67 | 0.20 | 0.00 | 0.00 | 1.00 | 2.72 | 7 |
| $P_3$ | 0.00 | 0.25 | 0.60 | 0.00 | 0.00 | 0.67 | 0.20 | 0.00 | 0.00 | 1.00 | 2.72 | 7 |
| $P_4$ | 0.00 | 0.25 | 0.60 | 0.00 | 0.00 | 0.33 | 0.20 | 0.25 | 0.00 | 1.00 | 2.63 | 9 |
| $P_5$ | 0.00 | 0.25 | 0.60 | 0.25 | 0.00 | 0.33 | 0.20 | 0.25 | 0.00 | 0.00 | 1.88 | 22 |
| $P_6$ | 0.00 | 0.25 | 0.60 | 0.00 | 0.00 | 0.67 | 0.20 | 0.25 | 0.00 | 0.00 | 1.97 | 15 |
| $P_7$ | 0.00 | 0.25 | 0.60 | 0.25 | 0.00 | 0.67 | 0.20 | 0.00 | 0.00 | 0.00 | 1.97 | 15 |
| $P_8$ | 0.00 | 0.50 | 0.60 | 0.25 | 0.00 | 0.67 | 0.20 | 0.25 | 0.33 | 0.00 | 2.8 | 6 |
| $P_9$ | 0.00 | 0.50 | 0.60 | 0.00 | 0.00 | 0.67 | 0.20 | 0.25 | 0.00 | 0.00 | 2.22 | 13 |
| $P_{10}$ | 0.00 | 0.25 | 0.60 | 0.00 | 0.00 | 0.67 | 0.20 | 0.00 | 0.00 | 0.00 | 1.72 | 25 |
| $P_{11}$ | 0.00 | 0.25 | 0.60 | 0.25 | 0.00 | 0.67 | 0.20 | 0.00 | 0.00 | 0.00 | 1.97 | 15 |
| $P_{12}$ | 0.00 | 0.25 | 0.60 | 0.25 | 0.00 | 0.33 | 0.20 | 0.00 | 0.33 | 0.00 | 1.96 | 21 |
| $P_{13}$ | 0.00 | 0.25 | 0.60 | 0.25 | 0.00 | 0.67 | 0.20 | 0.00 | 0.00 | 0.00 | 1.97 | 15 |
| $P_{14}$ | 0.00 | 0.25 | 0.60 | 0.25 | 0.00 | 0.33 | 0.20 | 0.25 | 0.67 | 1.00 | 3.55 | 1 |
| $P_{15}$ | 0.00 | 0.25 | 0.60 | 0.00 | 0.00 | 0.67 | 0.20 | 0.25 | 0.00 | 1.00 | 2.97 | 4 |
| $P_{16}$ | 0.00 | 0.25 | 0.60 | 0.00 | 0.00 | 0.67 | 0.20 | 0.25 | 0.00 | 0.00 | 1.97 | 15 |
| $P_{17}$ | 0.00 | 0.25 | 0.60 | 0.25 | 0.00 | 0.67 | 0.20 | 0.25 | 0.00 | 1.00 | 3.22 | 3 |
| $P_{18}$ | 0.00 | 0.25 | 0.60 | 0.25 | 0.00 | 0.67 | 0.20 | 0.50 | 0.00 | 1.00 | 3.47 | 2 |
| $P_{19}$ | 0.00 | 0.25 | 0.60 | 0.00 | 0.00 | 0.33 | 0.20 | 0.25 | 0.00 | 1.00 | 2.63 | 9 |
| $P_{20}$ | 0.00 | 0.25 | 0.60 | 0.00 | 0.00 | 0.33 | 0.20 | 0.00 | 0.00 | 0.00 | 1.38 | 27 |
| $P_{21}$ | 0.00 | 0.25 | 0.60 | 0.00 | 0.00 | 0.67 | 0.20 | 0.25 | 0.00 | 0.00 | 1.97 | 15 |
| $P_{22}$ | 0.00 | 0.25 | 0.60 | 0.25 | 0.00 | 0.67 | 0.20 | 0.25 | 0.00 | 0.00 | 2.22 | 13 |
| $P_{23}$ | 0.00 | 0.25 | 0.60 | 0.25 | 0.00 | 0.67 | 0.00 | 0.50 | 0.00 | 0.00 | 2.27 | 12 |
| $P_{24}$ | 0.00 | 0.25 | 0.60 | 0.00 | 0.00 | 0.33 | 0.20 | 0.25 | 0.00 | 0.00 | 1.63 | 26 |
| $P_{25}$ | 0.00 | 0.25 | 0.60 | 0.00 | 0.00 | 0.33 | 0.20 | 0.00 | 0.00 | 1.00 | 2.38 | 11 |
| $P_{26}$ | 0.00 | 0.25 | 0.60 | 0.00 | 0.00 | 0.33 | 0.20 | 0.00 | 0.00 | 0.00 | 1.38 | 27 |
| $P_{27}$ | 0.00 | 0.25 | 0.60 | 0.25 | 0.00 | 0.67 | 0.00 | 0.00 | 0.00 | 0.00 | 1.77 | 24 |
| $P_{28}$ | 0.00 | 0.25 | 0.60 | 0.00 | 0.00 | 0.67 | 0.20 | 0.25 | 0.00 | 1.00 | 2.97 | 4 |
| Average | 0.00 | 0.28 | 0.59 | 0.13 | 0.00 | 0.57 | 0.21 | 0.18 | 0.09 | 0.37 | 2.28 | / |
| Grade | 9 | 4 | 1 | 7 | 9 | 2 | 5 | 6 | 8 | 3 | / | / |

evaluation $X_9$, policy $P_{14}$ lacks regional characteristics, and does not formulate the policy flexibly according to the specific economic development of the region, making it difficult to judge whether or not the policy is in line with the actual situation and whether or not it can be implemented; and policy citation $X_{10}$ lacks the basis for policy formulation, and does not clearly point out the basis for policy formulation, and does not specify the policy citation $X_{10}$. The policy citation $X_{10}$ lacks the basis for policy formulation, and does not clearly indicate whether the policy formulation implements the relevant national regulations and spirit, making the policy program lack of scientific and persuasive power; moreover, in terms of policy means $X_8$, compared with most other policies, $P_{19}$ only adopts three initiatives of organization and coordination, supervision and evaluation, and publicity and guidance, and lacks the content of the rule of law, which indicates that the relevant safeguards are still unsound, and there is a lack of legal constraints and incentives, and the system building is also lacking in the content. constraints and incentives, and the institutional construction is not perfect. As far as $P_{18}$ and $P_{25}$ are concerned, the concavity indices of the two policies are the same after the assignment of each level of variables, and specific analysis shows that the concavity indices in the six aspects of policy nature $X_1$, policy content $X_4$, policy function

$X_5$, policy means $X_8$, policy evaluation $X_9$ and policy citation $X_{10}$ are all 0, and the degree of completeness is more prominent; at the same time, compared with the majority of other policies, $P_{18}$ and $P_{25}$ not only clarify the short-term development plans, but also take into account the medium-term development goals. Therefore, the order of policy improvement in $P_{19}$ is "$X_3$ policy object - $X_9$ policy evaluation - $X_{10}$ policy citation - $X_8$ policy means - $X_6$ policy time", and it should be further clarified that the path of policy improvement is not the only one, and it can be adjusted according to the actual situation in Hunan Province.

## 5. Conclusions and policy implications

### 5.1 Conclusions and suggestions

The Inter-provincial Government Services Policy enhances data sharing and collaboration among government agencies, enabling more informed decision-making and effective policy formulation. And it facilitates seamless integration and interoperability of government services across different regions, ensuring a cohesive experience for citizens and businesses. This paper uses the text mining method and PMC index model to quantitatively evaluate 28 texts of the Inter-provincial Government Services Policy, with a view to providing practical support for policy optimization. The main conclusions are as below.

1. The scope of application of the policy is relatively narrow, especially the policy area. Of the 28 policies, only the economic, political and scientific and technological fields are covered, with a small minority application in the environmental field.

2. The timeframe of the policy text mainly involves the long-term, medium-term and short-term. Short-term goals are usually more specific and operational and can help governments to better plan and allocate resources. Medium-term goals can help the Government to better measure and keep track of its progress, as well as make timely adjustments and improvements. Long-term goals require the Government to have a long-term vision and strong determination to provide direction and guidance for future development. Of the 28 policies, only 9 addressed medium-term goals, while the remaining 19 focused only on short-term plans, and all neglected long-term deployment. To some extent, this has led to a lack of foresight and sustainability in interprovincial government service policies.

3. The Inter-provincial Government Services policy as an emerging policy can be in the policy means, policy evaluation and policy citation three aspects of efforts to improve the scientific nature of the policy. In terms of policy means,16 policies do not specify the safeguards of relevant laws, regulations and systems, which will make the Inter-provincial Government Services incompatible documents and standards cannot be cleaned up and revised in a timely manner. As for policy evaluation, local policies are able to implement the guiding opinions of the central policy, but are lack of specific, implementable and distinctive programmers based on different local characteristics and development situations. As well, citations are less likely to appear in policy. There are 10 policies with insufficient basis to judge the viability of the policy.

Based on the above conclusion, the conclusions for policy optimization are proposed below.

1. Expanding the scope of application of government policies. In addition to the fields of economics, science and politics, the policy can also take into account the need for cross-location law enforcement in the field of ecological environmental protection, and actively

explore new areas of application. By expanding the areas of policy implementation, policies can benefit people's livelihoods in all aspects.

2. Focusing on the long-term effect of the policy. In the future, we should speed up the formulation of long-term goals. It is recommended that the central government should expeditiously issue a macroscopic and authoritative overall plan and constructive opinions to regulate and guide the development of Inter-provincial Government Services, so as to guide other provinces and municipalities to expeditiously issue corresponding policies according to the actual development situation of each region, and make systematic arrangements for the overall requirements, key tasks, and safeguards to avoid the wastage of resources and undue consequences due to the lack of a basis for blind planning.

3. Improving the quality of the government's policy making. The Inter-provincial Government Services Policy is still in the stage of exploration and development, and the policy makers need to continuously make adjustments according to the feedback information from the implementation. Efforts can be made in three areas, policy instruments, policy evaluation and policy citation, to form a virtuous cycle of policy making and policy implementation and realize the two-way promotion, so as to enhance the practical effectiveness of the Inter-provincial Government Services. a) Policy instruments: We need to strengthen institutional mechanisms and collaborative governance to promote digital government. b) Policy evaluation: Local governments need to identify shortcomings in the level of informatization and the degree of development of digital infrastructure. On the basis of making up for the shortcomings, we should formulate development programs that suit their local characteristics. c) Policy citation: It is necessary to clarify the guiding ideology of the policy and the requirements of society for building a service-oriented government that satisfies the people, so as to ensure that the policy is based on sufficient grounds and is in line with the actual situation.

## 5.2 Further discussion

By analyzing the text of China's inter-provincial general policy from 2020 to 2023, this paper quantitatively evaluates the policy as a whole, and obtains some useful conclusions and enlightenments. However, there are still some deficiencies and areas that can be improved in this paper. And it is difficult to conduct a more comprehensive analysis of the actual effect due to the insufficient sample size due to the short implementation time of the policy. In addition, the research method of this paper only explores the content of the policy text, and does not study the effects brought about by the implementation of the policy. We will consider combining text analysis and quantitative empirical research in the future, in order to provide a more comprehensive and scientific method and empirical reference for the quantitative evaluation of the policy.

## Supporting information

**S1 File.**
(DOCX)

## Author Contributions

**Conceptualization:** Rong-qing Geng, Jian Wu.

**Data curation:** Rong-qing Geng, Jian Wu.

**Formal analysis:** Rong-qing Geng, Jian Wu.

**Funding acquisition:** Rong-qing Geng, Jian Wu.

**Investigation:** Rong-qing Geng, Jian Wu.

**Methodology:** Rong-qing Geng, Jian Wu.

**Project administration:** Rong-qing Geng, Jian Wu.

**Resources:** Rong-qing Geng, Jian Wu.

**Software:** Rong-qing Geng, Jian Wu.

**Supervision:** Rong-qing Geng, Jian Wu.

**Validation:** Rong-qing Geng, Jian Wu.

**Visualization:** Rong-qing Geng, Jian Wu.

**Writing – original draft:** Rong-qing Geng, Jian Wu.

**Writing – review & editing:** Rong-qing Geng, Jian Wu.

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
