## [Decision Letter · Decision Letter 0]

11 Jun 2024

PONE-D-24-19514Assess the efficacy of China’s Inter-provincial Government Services Policy: A Quantitative Evaluation Based on PMC-Index ModelPLOS ONE

Dear Dr. Wu,

Thank you for submitting your manuscript to PLOS ONE. After careful consideration, we feel that it has merit but does not fully meet PLOS ONE’s publication criteria as it currently stands. Therefore, we invite you to submit a revised version of the manuscript that addresses the points raised during the review process.

We look forward to receiving your revised manuscript.

Kind regards,

Pengpeng Ye

Academic Editor

PLOS ONE

Journal Requirements:

"This work was supported by National Social Science Foundation of China (No. 23CSH057)，Hebei Natural Science Foundation (No. G2023203015).The funders had no role in study design, data collection and analysis, decision to publish, or preparation of the manuscript."

Reviewers' comments:

Reviewer's Responses to Questions

**Comments to the Author**

1. Is the manuscript technically sound, and do the data support the conclusions?

Reviewer #1: Yes

Reviewer #2: Partly

Reviewer #3: Partly

2. Has the statistical analysis been performed appropriately and rigorously? 

Reviewer #1: Yes

Reviewer #2: No

Reviewer #3: I Don't Know

3. Have the authors made all data underlying the findings in their manuscript fully available?

Reviewer #1: Yes

Reviewer #2: No

Reviewer #3: Yes

4. Is the manuscript presented in an intelligible fashion and written in standard English?

Reviewer #1: Yes

Reviewer #2: No

Reviewer #3: Yes

5. Review Comments to the Author

Reviewer #1: ABSTRACT

The abstract effectively introduces a study evaluating China's Inter-provincial Government Services Policy using a quantitative approach based on the PMC model. It succinctly presents the findings regarding policy design, consistency, and identified weaknesses, along with proposed optimization recommendations. However, it could benefit from clearer language and organization to enhance readability and impact.For instance, the author could consider restructuring the abstract to begin with a concise statement of the research aim or problem addressed by the Inter-provincial Government Services Policy. Then, provide a brief overview of the methodology employed, followed by a clear presentation of the key findings and their implications. Additionally, using simpler language and avoiding jargon can improve accessibility for readers unfamiliar with the PMC model or policy evaluation terminology.

INTRODUCTION

The introduction could benefit from clearer organization and language to improve readability. For instance, dividing the introduction into subsections such as Background, Significance of the Policy, Research Objective, and Highlights of the Paper would provide clearer structure. Using more concise language to convey key points would enhance readability and make the introduction more engaging.

LITERATURE REVIEW

The literature review provides a comprehensive overview of existing research on the Inter-provincial Government Services Policy and policy evaluation methods. To enhance clarity and readability, consider implementing the following suggestions:

a) The author can introduce clear subheadings for each subsection of the literature review to help readers navigate through different topics more easily. For example, the author could use subheadings like "Perspectives on Inter-provincial Government Services Policy" and "Methods of Policy Evaluation."

Recommendation:

Perspectives on Inter-provincial Government Services Policy

Collaborative Governance

Digital Transformation

Holistic Theoretical Approaches

b) The author needs to ensure that the language used is clear and accessible to readers from diverse backgrounds. Avoid overly technical terms or jargon unless it necessary.Recommendation:

Instead of: "The endogenous endowment of 'public-technical governance-overlapping action' of cross-domain governance of government services is refined by some scholars, who put forward three practice modes of cross-domain governance of government services: vertically embedded, internally originated hair style, and governance of government and society (enterprise), and constructed the development path of cross-domain governance of government services in the four dimensions of strategy, supply and demand, technology, and ecology development path."

Use: "Scholars refine the concept of 'public-technical governance-overlapping action' and propose three practice modes for cross-domain governance of government services: vertically embedded, internally originated, and governance of government and society."

RESEARCH DESIGN

a)The section lacks clear subheadings to delineate different aspects of the research design. Adding subsection headings such as "Research Sample and Methodology," "Policy Feature Recognition," "Construction of PMC Index Model," and "PMC Surface Drawing" would enhance readability and organization. Recommendation:

Before: "3.1 Research Sample and Methodology"

After: "3.1 Research Sample and Methodology: Selection Criteria and Data Collection"

b) The text contains lengthy explanations and repetitive phrases, which could be streamlined for brevity. For instance, instead of repeating phrases like "the high frequency of terms such as," the author could use more concise language to convey the same meaning.Recommendation:

Before: "From the perspective of policy guarantee mechanism, terms such as 'integration, management, acceptance, lead, docking coordination, support, guidance, coordination, supervision, and reliance' appear more frequently, indicating that accelerating the construction of Inter-provincial Government Services requires a variety of ways of coordination..."

After: "Policy guarantee mechanisms, including integration, management, acceptance, and coordination, are crucial for accelerating the construction of Inter-provincial Government Services."

c) The author needs to ensure smooth transition sentences between paragraphs can improve the flow of ideas. Connecting sentences that link each step of the research design process would provide readers with a clearer understanding of the methodology's progression. Recommendation:

Before: "3.2 Policy Feature Recognition. On the basis of obtaining the policy text..."

After: "3.2 Policy Feature Recognition: Following the acquisition of policy texts, the next step involves..."

d)Some sentences may be difficult for readers to comprehend. Simplifying complex phrases and using straightforward language would enhance clarity. Recommendation:

Before: "The visualization results show that keywords 'government', 'services', 'departments', 'sharing' and 'platforms' have a high centrality."

After: "The visualization results indicate that certain keywords, such as 'government' and 'services', hold significant centrality within the semantic network."

e) While the text references tables and figures, it would be beneficial to integrate them more seamlessly into the narrative. Directly referring to specific figures in the text and explaining their relevance to the discussion would enhance understanding.Recommendation:

Before: "The PMC surface figure is a symmetric surface based on the values of nine first-level indexes, which can show the pros and disadvantages of various policies, so as to better evaluate Inter-provincial Government Services policies formulated by the Chinese government."

After: "The PMC surface figure (Figure 1) displays a symmetric surface representing the values of nine first-level indexes, providing insights into the strengths and weaknesses of Inter-provincial Government Services policies."

RESULT AND COMPARATION ANALYSIS

a)The section title "Results and Comparative Analysis" could be clearer if it explicitly mentions what is being analyzed. Also, consider breaking down the content into smaller subsections for easier navigation and comprehension. Recommendation:

Before: "4. Results and Comparation analysis"

After: "4. Evaluation and Comparative Analysis of Inter-provincial Government Services Policies"

b) Some sentences contain redundant phrases or repetitive information. Try to convey the same meaning with fewer words to improve readability and efficiency. Recommendation:

Before: "From the calculation results of the PMC index of the 28 policies (see Table 6), the PMC index scores range from [6.45 to 8.62], with a mean value of 7.71, and 15, 16, and 19 are acceptable levels, while the rest of the policies are excellent levels, and there are no bad and perfect policies. perfect level policies."

After: "The PMC index scores of the 28 policies ranged from 6.45 to 8.62, with a mean value of 7.71. Notably, policies 15, 16, and 19 attained acceptable levels, while the others achieved excellence."

c)The author need to introduce smoother transitions between subsections to improve the flow of the analysis. Clear transitions help guide the reader through the logical progression of ideas. Recommendation:

Before: "4.2.1 Characterization of overall policies"

After: "4.2.1 Overview of Policies: Analyzing Overall Characteristics"

d) The author needs to simplify complex sentences and technical language to ensure clarity and accessibility for all readers. Recommendation:

Before: "Comparing the scores of each level of policy variables with the mean value, it is found that..."

After: "Upon comparing the scores of each policy variable with the mean value, it becomes evident that..."

e) The author needs to ensure that references to figures are clear and seamlessly integrated into the text. Figures should complement the analysis and be introduced in a way that enhances understanding.Recommendation:

Before: "In order to visualize the results of PMC index calculation, the first-level index scores of each policy can be transformed into a third-order matrix as shown in Eq.(5), and then the internal consistency level of each policy can be analyzed by observing the concavity and convexity of the surface."

After: "To visually represent the PMC index calculation results, we transform the first-level index scores of each policy into a third-order matrix (Eq. 5), facilitating analysis of each policy's internal consistency level through surface concavity and convexity."

CONCLUSION

The author needs to:

(a) Divide the conclusion into smaller paragraphs focusing on specific aspects, such as findings, recommendations, and future research directions.

(b) Ensure consistent use of terminology, such as "Inter-provincial Government Services Policy," throughout the conclusion.

(c) Refer to relevant figures, such as the PMC index calculation results, to support statements about policy effectiveness or areas for improvement.

Reviewer #2: 1. In the literature review, the contents of Section 2.1 are not strongly related to its title, Research on Inter-provincial Government Services Policy. The authors mainly illustrate collaborative governance, digital transformation, and holistic governance, with limited research reviews on Inter-provincial Government Services Policy.

2. What is the significance of the keyword network analysis in Section 3.2? How does it contribute to the construction of the PMC index model? Please make the rationale clearer.

3. In Section 3.3.4, PMC Surface Drawing, the text mentions three figures (Figures 2-4), but only Figure 2 is seen at the end of the paper. Please confirm whether the uploaded images are correct.

4. In the conclusion section, the author states that the main body of policy formulation is the government, and that experts, think tanks, and research institutions are not involved in policy formulation. How was this conclusion reached? How did the author obtain information on the composition of policy participants from text analysis and the PMC index model?

5. The article lacks discussions on the limitations.

6. The study uses the PMC index model to evaluate China’s Inter-provincial Government Service Policy, but the theoretical contribution and practical significance of the study are not well embodied.

Reviewer #3: This paper applies the PMC Index Model to quantitatively assess inter-provincial government services policies in China to explore the effectiveness of policy implementation, which indeed is a beneficial research direction. The literature review is clearly structured, advances critical analysis of existing research, and achieves positive results in its conclusions.

The introduction directly presents the research question and clearly lists the contributions of the study. However, I believe that the choice of research methods should be briefly described at the beginning, highlighting the systematic, holistic, and predictive features and advantages of the policy text evaluation method to smoothly transition into the subsequent research.

The paper's conclusions focus on proposing policy optimizations, which need to be more grounded in the research data and results. I perceive some issues with parts of the conclusions where the derivations from the data and data analysis seem less validating than verifying, and the depth of the conclusions appears somewhat lacking. I recommend that the author:

Precisely align the logic connection between data analysis results and conclusions.

Strengthen causal inferences to ensure all conclusions and recommendations are directly based on data analysis results.

The manuscript is clearly written and meets the norms of English academic writing, but there are still some inaccuracies in grammar and word choice that might affect reader comprehension. Further polishing of the language is needed to meet the standards of academic publication.

Overall, the article is suitable for publication in a journal.

6. PLOS authors have the option to publish the peer review history of their article (what does this mean?). If published, this will include your full peer review and any attached files.

Reviewer #1: **Yes: **FARAH ADILLA AB RAHMAN

Reviewer #2: No

Reviewer #3: No

---

## [Author Response · Author response to Decision Letter 0]

25 Jul 2024

Dear editors and reviewers:

We thank you very much for your helpful and insightful comments and suggestions on our manuscript entitled “Assess the efficacy of China’s Inter-provincial Government Services Policy: A Quantitative Evaluation Based on PMC-Index Model” .These comments and suggestions are very helpful for improving our paper. We studied them carefully and made some revisions according to the comments and suggestions of the associate editor and the reviewers. The main changes in the paper are marked in red color and the responses to the comments are as follows.

Reviewer #1: ABSTRACT

The abstract effectively introduces a study evaluating China's Inter-provincial Government Services Policy using a quantitative approach based on the PMC model. It succinctly presents the findings regarding policy design, consistency, and identified weaknesses, along with proposed optimization recommendations. However, it could benefit from clearer language and organization to enhance readability and impact. For instance, the author could consider restructuring the abstract to begin with a concise statement of the research aim or problem addressed by the Inter-provincial Government Services Policy. Then, provide a brief overview of the methodology employed, followed by a clear presentation of the key findings and their implications. Additionally, using simpler language and avoiding jargon can improve accessibility for readers unfamiliar with the PMC model or policy evaluation terminology. 

Response: We are very grateful to the reviewer for the positive comments. Based on the reviewers' suggestions, this paper has restructured the abstract. First, the author added a concise statement of the research aim at the beginning. Next, the author provided a brief overview of the methodology employed, followed by a clear presentation of the key findings and implications. At the same time, we have revised the paper in more concise language according to your request. All changes are shown in red. (See Abstract, Page 1).

INTRODUCTION

The introduction could benefit from clearer organization and language to improve readability. For instance, dividing the introduction into subsections such as Background, Significance of the Policy, Research Objective, and Highlights of the Paper would provide clearer structure. Using more concise language to convey key points would enhance readability and make the introduction more engaging.

Response: Thank you for your comments. Based on your suggestions, we have rearranged the introduction into background, policy implications, research objectives, and highlights. All changes are reflected in red. (See Introduction, Page 1-Page 3).

LITERATURE REVIEW

The literature review provides a comprehensive overview of existing research on the Inter-provincial Government Services Policy and policy evaluation methods. To enhance clarity and readability, consider implementing the following suggestions:

a) The author can introduce clear subheadings for each subsection of the literature review to help readers navigate through different topics more easily. For example, the author could use subheadings like "Perspectives on Inter-provincial Government Services Policy" and "Methods of Policy Evaluation."

Recommendation:

Perspectives on Inter-provincial Government Services Policy

Collaborative Governance

Digital Transformation

Holistic Theoretical Approaches

b) The author needs to ensure that the language used is clear and accessible to readers from diverse backgrounds. Avoid overly technical terms or jargon unless it necessary. Recommendation:

Instead of: "The endogenous endowment of 'public-technical governance-overlapping action' of cross-domain governance of government services is refined by some scholars, who put forward three practice modes of cross-domain governance of government services: vertically embedded, internally originated hair style, and governance of government and society (enterprise), and constructed the development path of cross-domain governance of government services in the four dimensions of strategy, supply and demand, technology, and ecology development path."

Use: "Scholars refine the concept of 'public-technical governance-overlapping action' and propose three practice modes for cross-domain governance of government services: vertically embedded, internally originated, and governance of government and society."

Response: We really appreciate the valuable advice. Firstly, we have added subheadings for the first subsection of the literature review. Secondly, we have trimmed and modified some of the statements to make them easier for readers of different backgrounds to understand. And the content is marked in red color. (See Section 2.1, Page 3- Page 4)

RESEARCH DESIGN

a)The section lacks clear subheadings to delineate different aspects of the research design. Adding subsection headings such as "Research Sample and Methodology," "Policy Feature Recognition," "Construction of PMC Index Model," and "PMC Surface Drawing" would enhance readability and organization. Recommendation:

Before: "3.1 Research Sample and Methodology"

After: "3.1 Research Sample and Methodology: Selection Criteria and Data Collection"

b) The text contains lengthy explanations and repetitive phrases, which could be streamlined for brevity. For instance, instead of repeating phrases like "the high frequency of terms such as," the author could use more concise language to convey the same meaning. Recommendation:

Before: "From the perspective of policy guarantee mechanism, terms such as 'integration, management, acceptance, lead, docking coordination, support, guidance, coordination, supervision, and reliance' appear more frequently, indicating that accelerating the construction of Inter-provincial Government Services requires a variety of ways of coordination..."

After: "Policy guarantee mechanisms, including integration, management, acceptance, and coordination, are crucial for accelerating the construction of Inter-provincial Government Services."

c) The author needs to ensure smooth transition sentences between paragraphs can improve the flow of ideas. Connecting sentences that link each step of the research design process would provide readers with a clearer understanding of the methodology's progression. Recommendation:

Before: "3.2 Policy Feature Recognition. On the basis of obtaining the policy text..."

After: "3.2 Policy Feature Recognition: Following the acquisition of policy texts, the next step involves..."

d) Some sentences may be difficult for readers to comprehend. Simplifying complex phrases and using straightforward language would enhance clarity. Recommendation:

Before: "The visualization results show that keywords 'government', 'services', 'departments', 'sharing' and 'platforms' have a high centrality."

After: "The visualization results indicate that certain keywords, such as 'government' and 'services', hold significant centrality within the semantic network."

e) While the text references tables and figures, it would be beneficial to integrate them more seamlessly into the narrative. Directly referring to specific figures in the text and explaining their relevance to the discussion would enhance understanding. Recommendation:

Before: "The PMC surface figure is a symmetric surface based on the values of nine first-level indexes, which can show the pros and disadvantages of various policies, so as to better evaluate Inter-provincial Government Services policies formulated by the Chinese government."

After: "The PMC surface figure (Figure 1) displays a symmetric surface representing the values of nine first-level indexes, providing insights into the strengths and weaknesses of Inter-provincial Government Services policies."

Response: Thanks for the valuable suggestions. We have made changes to the corresponding sections according to your request, as follows: First, we modified the subheadings utilized in the research design section to make it clearer what follows.(See Section 3.1 and 3.2, Page 5 and Page 7).Second, we have deleted lengthy explanations and repetitive phrases, and replaced them with more concise expressions (See Section 3.2, Page 7).Next, in section 3.1 and 3.2, we have added transitional sentences between paragraphs to make the process of research design clearer to the readers (See Section 3.1 and Section 3.2,Page 5 and Page 7).Additionally, we've simplified complex phrases and used straightforward language for greater clarity (See Section 3.2,Page 9). Finally, we make direct references to specific charts in the text and explain their relevance to the discussion, allowing the table to fit more seamlessly into the narrative (See Section 3.3.4, Page 12). Specific changes can be found in red in the paper.

RESULT AND COMPARATION ANALYSIS

a)The section title "Results and Comparative Analysis" could be clearer if it explicitly mentions what is being analyzed. Also, consider breaking down the content into smaller subsections for easier navigation and comprehension. Recommendation:

Before: "4. Results and Comparation analysis"

After: "4. Evaluation and Comparative Analysis of Inter-provincial Government Services Policies"

b) Some sentences contain redundant phrases or repetitive information. Try to convey the same meaning with fewer words to improve readability and efficiency. Recommendation:

Before: "From the calculation results of the PMC index of the 28 policies (see Table 6), the PMC index scores range from [6.45 to 8.62], with a mean value of 7.71, and 15, 16, and 19 are acceptable levels, while the rest of the policies are excellent levels, and there are no bad and perfect policies. perfect level policies."

After: "The PMC index scores of the 28 policies ranged from 6.45 to 8.62, with a mean value of 7.71. Notably, policies 15, 16, and 19 attained acceptable levels, while the others achieved excellence."

c)The author need to introduce smoother transitions between subsections to improve the flow of the analysis. Clear transitions help guide the reader through the logical progression of ideas. Recommendation:

Before: "4.2.1 Characterization of overall policies"

After: "4.2.1 Overview of Policies: Analyzing Overall Characteristics"

d) The author needs to simplify complex sentences and technical language to ensure clarity and accessibility for all readers. Recommendation:

Before: "Comparing the scores of each level of policy variables with the mean value, it is found that..."

After: "Upon comparing the scores of each policy variable with the mean value, it becomes evident that..."

e) The author needs to ensure that references to figures are clear and seamlessly integrated into the text. Figures should complement the analysis and be introduced in a way that enhances understanding. Recommendation:

Before: "In order to visualize the results of PMC index calculation, the first-level index scores of each policy can be transformed into a third-order matrix as shown in Eq.(5), and then the internal consistency level of each policy can be analyzed by observing the concavity and convexity of the surface."

After: "To visually represent the PMC index calculation results, we transform the first-level index scores of each policy into a third-order matrix (Eq.5), facilitating analysis of each policy's internal consistency level through surface concavity and convexity."

Response: Thanks for these helpful suggestions. We've made changes to the content in line with the detailed suggestions you have given. First, we revised the title of Section 4 and subdivided the content of the title of 4.2.1 into smaller subsections so that it clearly mentions what is being analyzed.(See Section 4 and Section 4.2.1,Page 12 and Page 14).Second, we eliminated sentences containing redundant phrases or repetitive information to convey the same meaning in fewer words.(See Section 4.2.1 and Section 4.3.1,Page 14 and Page 15).After that, we revised the title of 4.2.1 to allow for smoother transitions between subsections so that the reader understands the logical progression of ideas.(See Section 4.2.1,Page 14). Then, we simplified complex sentences and technical language to ensure clarity and accessibility for all readers (See Section 4.3.1, Page 15).Finally, We modified the expression of the sentences to ensure that references to figures are clear and seamlessly integrated into the text.(See Section 4.2.2, Page 14).All changes are highlighted in red.

CONCLUSION

The author needs to:

(a) Divide the conclusion into smaller paragraphs focusing on specific aspects, such as findings, recommendations, and future research directions.

(b) Ensure consistent use of terminology, such as "Inter-provincial Government Services Policy," throughout the conclusion.

(c) Refer to relevant figures, such as the PMC index calculation results, to support statements about policy effectiveness or areas for improvement.

Response: Thanks for the valuable suggestion. (a) We have revised the conclusion of the paper according to your requirements, and marked the corresponding revisions in red, which part of the article is specified. (b) We have modified some expressions to ensure that terminology is used consistently throughout the conclusions. (c) In the conclusion, we cite figures to support statements about the effectiveness of the policy or areas for improvement. (See Section 5.1, Page 18-Page 20).The changes have been highlighted in red.

Reviewer #2: 1. In the literature review, the contents of Section 2.1 are not strongly related to its title, Research on Inter-provincial Government Services Policy. The authors mainly illustrate collaborative governance, digital transformation, and holistic governance, with limited research reviews on Inter-provincial Government Services Policy.

Response: We greatly appreciate that the reviewer has pointed this out. We have added a synthesis of some studies on Inter-provincial Government Services Policy and made appropriate changes and adjustments to the content.(See Section 2,Page 3-Page 4).

2. What is the significance of the keyword network analysis in Section 3.2? How does it contribute to the construction of the PMC index model? Please make the rationale clearer.

Response: According to the reviewer’s suggestion, we added the significance of keyword network analysis and showed how it can help in constructing PMC index models. (See section 3.2,Page 9).

3. In Section 3.3.4, PMC Surface Drawing, the text mentions three figures (Figures 2-4), but only Figure 2 is seen at the end of the paper. Please confirm whether the uploaded images are correct.

Response: Please forgive us for leaving out two pictures. We have added two figures to the document.(See Figure 2 and Figure 3).

4.In the conclusion section, the author states that the main body of policy formulation is the government, and that experts, think tanks, and research institutions are not involved in policy formulation. How was this conclusion reached? How did the author obtain information on the composition of policy participants from text analysis and the PMC index model?

Response: Thanks for pointing out the issues. We are very sorry not to explain how the conclusions were reached, but in fact we have come to the composition of policy actors through statistical analysis, i.e., the main body of policy-making is the government, experts, think tanks and research institutions. Among them, experts, think tanks and research institutions are less involved in policy- making. To avoid ambiguity, we have removed the corresponding elements from the conclusions and reorganized them to make them more revealing.(See Section 5.1,Page 19).

5.The article lacks discussions on the limitations.

Response: Thanks for the valuable suggestion. We add a discussion of limitations at the end of the article.(See Section 5.2,Page 20).

6. The study uses the PMC index model to evaluate China’s Inter-provincial Government Service Policy, but the theoretical contribution and practical significance of the study are not well embodied.

Response: Thank you for your constructive comments. We apologize for not clarifying the theoretical contribution and practical implications of this article. We have rearranged the article, and the theoretical contributions and practical sign

---

## [Decision Letter · Decision Letter 1]

2 Sep 2024

Assess the efficacy of China’s Inter-provincial Government Services Policy: A Quantitative Evaluation Based on PMC-Index Model

PONE-D-24-19514R1

Dear Dr. Wu,

We’re pleased to inform you that your manuscript has been judged scientifically suitable for publication and will be formally accepted for publication once it meets all outstanding technical requirements.

Kind regards,

Pengpeng Ye

Academic Editor

PLOS ONE

Additional Editor Comments (optional):

Reviewers' comments:

Reviewer's Responses to Questions

**Comments to the Author**

1. If the authors have adequately addressed your comments raised in a previous round of review and you feel that this manuscript is now acceptable for publication, you may indicate that here to bypass the “Comments to the Author” section, enter your conflict of interest statement in the “Confidential to Editor” section, and submit your "Accept" recommendation.

Reviewer #1: All comments have been addressed

Reviewer #3: All comments have been addressed

2. Is the manuscript technically sound, and do the data support the conclusions?

Reviewer #1: Yes

Reviewer #3: Partly

3. Has the statistical analysis been performed appropriately and rigorously? 

Reviewer #1: Yes

Reviewer #3: Yes

4. Have the authors made all data underlying the findings in their manuscript fully available?

Reviewer #1: Yes

Reviewer #3: No

5. Is the manuscript presented in an intelligible fashion and written in standard English?

Reviewer #1: Yes

Reviewer #3: Yes

6. Review Comments to the Author

Reviewer #1: Dear Dr. Jian Wu,

I want to extend my sincere appreciation for the thoughtful revisions you've made based on my earlier suggestions. The restructuring of the abstract, particularly the addition of a concise statement of the research aim, has significantly improved the clarity and impact of your study. Additionally, the reorganization of the introduction and literature review, along with the enhancements in language simplicity, have greatly contributed to the readability of your work.

Your attention to detail, such as integrating subheadings in the research design and simplifying complex phrases, has made the methodology and findings more accessible to a broader audience. The adjustments in the results and comparative analysis sections, including the refined transitions and references to figures, have also strengthened the overall flow and coherence of your paper.

It has been a pleasure to review your work, and I am confident that these changes will greatly benefit your readers. I wish you all the best in your future research endeavors and look forward to seeing your continued contributions to the field.

Best regards,

Dr. Farah Adilla Ab Rahman

Reviewer #3: (No Response)

7. PLOS authors have the option to publish the peer review history of their article (what does this mean?). If published, this will include your full peer review and any attached files.

Reviewer #1: **Yes: **Dr. Farah Adilla Ab Rahman

Reviewer #3: No

---

## [Editor Report · Acceptance letter]

13 Sep 2024

PONE-D-24-19514R1 

PLOS ONE

Dear Dr. Wu, 

I'm pleased to inform you that your manuscript has been deemed suitable for publication in PLOS ONE. Congratulations! Your manuscript is now being handed over to our production team.

Kind regards, 

on behalf of

Dr. Pengpeng Ye 

Academic Editor

PLOS ONE